# OpenReview forum: "MMDT: Decoding the Trustworthiness and Safety of Multimodal Foundation Models"
_ICLR.cc/2025/Conference — ICLR 2025 Poster_

### Official Review · Reviewer_mkAr · 2024-10-27

**Soundness:** 3
**Presentation:** 3
**Contribution:** 3
**Rating:** 6
**Confidence:** 3

**Summary:**

The paper introduces Multimodal DecodingTrust (MMDT) to provide a comprehensive evaluation of safety and trustworthiness across multiple dimensions: safety, hallucination, fairness, privacy, adversarial robustness, and out-of-distribution generalization. MMDT includes diverse evaluation scenarios and red teaming algorithms, creating challenging data for each evaluation task to rigorously assess these perspectives.

**Strengths:**

The paper provides a comprehensive evaluation framework to evaluate Multimodal Foundation Models (MMFMs).

The evaluation using MMDT reveals specific vulnerabilities and areas of improvement in multimodal models, providing actionable insights for researchers and developers.

**Weaknesses:**

As a benchmark paper, I did not find many unique issues of MMFM compared to the general LLM or foundation models. I appreciate the large workload for benchmarking, but this is more like an experimental report than a solid research paper. For example, there are limited analytical components or theoretical insights.

**Questions:**

Can you emphasize some unique discoveries from your evaluation process? For example, something that is not straightforward, or something happens in MMFM but not in pure LLM?

---

> ### Author Response · Authors · 2024-11-22
> **Response to Reviewer mkAr**
>
> We sincerely thank the reviewer for their detailed feedback and for acknowledging the comprehensive scope of our evaluation framework. Below, we address the concerns and elaborate on the unique contributions of our work.
>
> > **Q1.** Unique issues of MMFMs compared to general LLMs
>
> Thank you for your valuable feedback. We acknowledge that MMFMs share certain vulnerabilities with LLMs. However, our work identifies and emphasizes several unique challenges specific to MMFMs that are not observed in pure LLMs. Below are some unique discoveries from our evaluations that underline these challenges:
>
> 1. Cross-Modal Vulnerabilities in I2T Models: In image-to-text models, we find that attacks in one modality (e.g., adversarial images) can lead to severe failures in text predictions, such as unsafe behaviors, even when the benign text inputs would yield safe outputs. This cross-modal vulnerability is unique to MMFMs due to their multi-modal input alignment mechanisms.
>
> 1. Architecture-Dependent Vulnerabilities in T2I Models: Text-to-image (T2I) models, often based on guided diffusion models, exhibit unique vulnerabilities not present in transformer-based text-only LLMs. For example, adversarially perturbed prompts in MMDT often lead to irrelevant content generation, highlighting distinct risks that do not appear in text-only models.
>
> 1. Imbalance in Modality Contributions: We observe that I2T MMFMs often place disproportionate emphasis on textual or visual information. This imbalance leads to hallucination failures specific to MMFMs, such as incorrect attributions of objects or actions to visual content. Such failures do not appear in pure text-based models.
>
> We will incorporate these findings into the main text and conclusion of the revised manuscript to better emphasize the distinct challenges of MMFMs.
>
> > **Q2.** Limited analytical components or theoretical insights
>
> Thank you for your feedback. While MMDT primarily serves as an evaluation framework, it also provides analytical insights into MMFMs' behaviors across multiple dimensions. We highlight the following general conclusions drawn from our evaluations:
>
> 1. Cross-Modality Vulnerabilities: MMDT reveals how vulnerabilities in one modality can influence the other. For instance, adversarial perturbations in text prompts can lead T2I models to generate irrelevant images, while adversarial images in I2T models cause incorrect textual descriptions. These cross-modality dependencies are now present in pure LLMs.
>
> 1. Cross-Perspective Correlations. Through comprehensive evaluations, we observe correlations between certain risk perspectives. For instance, adversarial robustness and OOD robustness exhibit closely related trends across different MMFMs, suggesting potential shared vulnerabilities and mitigation strategies.
>
> 1. Trustworthiness Trade-Offs. Our findings indicate that improving fairness in certain models often comes at the cost of safety or hallucination performance, highlighting the inherent trade-offs in achieving comprehensive trustworthiness in MMFMs.
>
> > **Q3.** Can you emphasize some unique discoveries from your evaluation process? For example, something that is not straightforward, or something happens in MMFM but not in pure LLM?
>
> Thank you for your suggestion. Below are some notable discoveries unique to MMFMs, which we will elaborate on in the revised version:
>
> 1. Distraction Hallucinations in I2T Models: When presented with irrelevant but visually prominent objects in an image, I2T models frequently prioritize these distractions over the main subject of the image, leading to hallucinated descriptions. This behavior does not occur in pure LLMs, which do not process visual inputs.
>
> 1. Attribute Amplification in T2I Models: When tasked with generating images based on text prompts describing sensitive attributes (e.g., gender, race), T2I models tend to amplify stereotypes, resulting in highly biased or unsafe outputs. This amplification in image generation is unique to T2I models, especially diffusion models, which have significant differences from transformer-based LLMs.
>
> 1. Sensitivity to Cross-Modality Noise: We found that combining benign text with adversarial images (or vice versa) can lead to significant issues in output contents. For instance, I2T models often generate hallucinated visual elements or unsafe responses when the image is slightly perturbed, even when the original prompt is clear.
>
> 1. Distinct Privacy Risks in Images: Unlike LLMs, MMFMs exhibit privacy risks unique to the image domain, such as the unintended leakage of sensitive visual details in outputs (e.g., recognizable faces or locations) when queried with adversarial prompts.

---

> > ### Comment · Reviewer_mkAr · 2024-11-22
> >
> > Thank you for your response, and I really appreciate your hard work. Some of my concerns are solved, and the score will be updated.

---

### Official Review · Reviewer_SJeB · 2024-10-31

**Soundness:** 3
**Presentation:** 3
**Contribution:** 3
**Rating:** 6
**Confidence:** 4

**Summary:**

This paper introduces MMDT, a unified platform for evaluating the safety and trustworthiness of MMFMs. Unlike existing benchmarks, MMDT assesses models across multiple dimensions, including safety, hallucination, fairness/bias, privacy, adversarial robustness, and OOD generalization. The platform includes diverse evaluation scenarios and red teaming algorithms to generate challenging data. Evaluations conducted using MMDT reveal key vulnerabilities and areas for improvement in multimodal models, establishing MMDT as a foundational tool for advancing safer and more reliable MMFMs.

**Strengths:**

Strengths:

+ This work offers MMDT, a unified platform for evaluating the safety and trustworthiness of MMFMs.
+ Comprehensive and thorough assessment.
+ Revealed the security risk of the MMFMs.

**Weaknesses:**

Weaknesses:

- No obvious weaknesses.

**Questions:**

Questions:

I am pleased to see that the authors have provided a comprehensive and in-depth benchmark evaluation of MMFMs in terms of safety, hallucination, fairness, and privacy, particularly including mainstream commercial models such as GPT-4 in the evaluation. Additionally, the availability of detailed code, data, and case examples is valuable, allowing readers to conduct further research and extensions. However, I have a few additional questions I would like to discuss with the authors.


- Q1: Although the authors provide a comprehensive and rigorous assessment of MMFMs, especially through the design of red team models for each evaluation objective, it would be beneficial if the authors could also briefly discuss blue team design. Would the authors consider adding a discussion on future blue team approaches in the manuscript?

- Q2: Have the authors examined the security threats of multimodal inputs to MMFMs? For instance, [R1] presents a multimodal red team model for MMFMs; addressing this would expand the assessment's scope.

[R1] Liu Y, Cai C, Zhang X, et al. Arondight: Red Teaming Large Vision Language Models with Auto-generated Multi-modal Jailbreak Prompts[C]//Proceedings of the 32nd ACM International Conference on Multimedia. 2024: 3578-3586.

- Q3: Could the authors elaborate on the differences and connections between the red team model introduced in this paper and existing work? Highlighting these distinctions would improve clarity.

- Q4: For dynamically evolving MMFMs, how does the proposed red team model adapt to ongoing changes? It would be helpful if the authors could address whether this adaptability has been considered in the current model.

- Q5: Finally, have the authors accounted for the diversity of test cases? Given that diversity is a crucial indicator of a red team model's effectiveness, an overview of this aspect would strengthen the evaluation.

---

> ### Comment · Reviewer_SJeB · 2024-11-21
> **Rebuttal?**
>
> I do not seem to have received any rebuttal from the authors.

---

> ### Author Response · Authors · 2024-11-22
> **Response to Reviewer SJeB (Part 1)**
>
> We thank the reviewer for the thoughtful feedback and for recognizing the strengths of our work, including MMDT's comprehensive evaluation framework, its inclusion of mainstream commercial models, and the provision of detailed code, data, and case examples. Below, we address each question and provide corresponding updates to strengthen the manuscript.
>
> > **Q1.** Brief discussion on blue team design
>
> Thank you for your valuable suggestion. We have expanded our discussion of blue team approaches for each trustworthiness perspective, detailed in Appendix C.3 of the revised paper. Key strategies include:
>
>  - Safety: Given that safety risks persist across all target models according to our evaluation, we recommend implementing more advanced mitigation strategies at various stages. During the training stage, utilizing Reinforcement Learning from Human Feedback (RLHF) with high-quality data based on a comprehensive taxonomy of risk categories is essential to reduce the risk of generating unsafe content. During the deployment stage, input and output-level guardrails, such as [1] and [2], could be adopted to detect and filter out unsafe content provided by users or generated by the model. Additionally, developing certified defenses against jailbreaking attacks can further mitigate these risks.
>
>  - Hallucination: Our findings in MMDT suggest that MMFMs tend to hallucinate primarily due to: 1) poor visual grounding; 2) imbalanced attention between textual and visual information; and 3) poor reasoning or instruction-following abilities. To address these issues, we can consider the following mitigation strategies: 1) utilize external tools to enhance visual grounding, as indicated in [3]; 2) adaptively calibrate attention to ensure a balanced focus on both textual and visual tokens; 3) employ supervised fine-tuning or preference tuning to reduce hallucination during training; and 4) leveraging external knowledge base for factual image retrieval to mitigate hallucination for text-to-image generation.
>
>  - Fairness: Since image generation models exhibit more severe bias issues compared to text generation models based on our observations, we advocate for the development of more effective bias mitigation strategies in the image domain. Addressing bias in image generation models is particularly challenging due to their increased complexity and the absence of automatic reward feedback. Therefore, leveraging human preference annotations and techniques such as DPO to enforce fairness in text-to-image models presents a promising direction for future work. Moreover, we emphasize the importance of pursuing fairness goals while avoiding overkill fairness that sacrifices historical and factual accuracy. Balancing these two objectives is particularly challenging, and we call for extensive research to address this complex issue.
>
>  - Privacy: Based on our evaluation results, we recommend employing privacy-preserving techniques during both the training and inference stages for MMFMs. During training, using differentially private learning algorithms or differentially private synthetic multimodal data could help alleviate concerns about privacy leakage. For inference, we suggest implementing scrubbing or anonymization on user-provided images and text to remove sensitive attributes. Additionally, MMFMs could incorporate privacy-aware instruction tuning and reject queries related to sensitive human attributes.
>
>  - Adversarial Robustness: According to our findings that adversarial examples generated by GCG and MMP has high transferability to other black-box models, we recommend to employ these algorithms to attack a wide range of open models to collect challenging data and mix the data into the training blend. These adversarial datasets will help improve model robustness vie trustworthiness fine-tuning.
>
>  - OOD Robustness: Given that superior in-distribution performance in MMDT typically leads to better out-of-distribution performance, we recommend further enhancing the models’ benign performance by increasing the training dataset quality and diversity. Additionally, collecting a diverse training dataset with various styles through data augmentation and incorporating diverse tasks, such as spatial reasoning and attribute recognition, can potentially improve the robustness of multi-modal models against different styles and tasks. Furthermore, we recommend multi-modal models to incorporate “I don’t know” option during training, enabling them to perform OOD detection by themselves rather than generating random answers.

---

> ### Author Response · Authors · 2024-11-22
> **Response to Reviewer SJeB (Part 2)**
>
> > **Q2.** Examining security threats of multimodal inputs
>
> Thank you for the insightful question. Our dataset is constructed and optimized specifically for multimodal inputs, incorporating diverse and challenging scenarios across six perspectives. Below are some representative examples:
>
> - Safety: We include scenarios where multimodal inputs such as harmful intent hidden in visual elements (e.g., offensive symbols or images combined with benign text) test the model's ability to detect unsafe content across modalities.
>
> - Hallucination: Multimodal prompts are designed to evaluate how well models handle visual-textual consistancies, such as presenting an image of two laptops with a textual query asking for "What would the color of the bottom laptop be if the red laptop and the white laptop were switched?"
>
> - Fairness: We develop a comprehensive testbed to evaluate cross-modal reasoning in models and uncover potential biases in the process. For example, we present images of job candidates from different racial backgrounds and ask the model to make hiring decisions based in text. This setup requires the model to jointly comprehend and reason across vision and language modalities.
>
> Additionally, while Arondight focuses solely on safety, our evaluation framework spans six trustworthiness perspectives, including hallucination, fairness, privacy, adversarial robustness, and OOD generalization. This broader scope ensures a more comprehensive assessment of MMFMs against challenging multimodal inputs.
>
> > **Q3.** Differences and connections between this red team model and existing work
>
> Thank you for the suggestion. We include a comparison in Table 52 summarizing the distinctions between our red team model and existing methods. Key differences include:
>
> 1. Multi-modality Coverage: While some existing works focus predominantly on text-to-image (T2I) or image-to-text (I2T) tasks, our approach balances the evaluation of both modalities, ensuring comprehensive coverage and cross-modal consistency.
>
> 2. Comprehensive Trustworthiness Evaluations: Unlike prior work, our red team model evaluates MMFMs across six trustworthiness perspectives, including privacy, fairness, and OOD generalization, in addition to safety and hallucination.
>
> 3. Scenario Diversity: MMDT incorporates novel red team-generated data for scenarios like harmful intent hidden in visual cues or multimodal context shifts, which are not covered in prior work.
>
> > **Q4.** Adaptability of the red team model for dynamically evolving MMFMs
>
> Thank you for this insightful question. Adaptability is a core consideration in our proposed red team model, ensuring its relevance for dynamically evolving MMFMs. Our approach includes:
>
> 1. Dynamic Data Generation: Our framework dynamically generates new data for trustworthiness evaluations, leveraging optimization-based methods to create challenging instances. This ensures that the evaluations remain rigorous even as MMFMs evolve.
>
> 1. Private Data for Future Evaluations: To avoid becoming obsolete through adversarial training by potential adversaries, newly generated red team data will be kept private and updated periodically. This approach maintains the platform's ability to evaluate future MMFMs effectively and prevents misuse.
>
> 1. Adaptability Across Models: For specific perspectives like adversarial robustness, our optimization algorithms can be seamlessly applied to more advanced MMFMs, enabling the generation of additional adversarial instances that address newly introduced vulnerabilities in ongoing models.
>
> These considerations are discussed in Appendix B of the revised manuscript to emphasize the adaptability of our approach.

---

> ### Author Response · Authors · 2024-11-22
> **Response to Reviewer SJeB (Part 3)**
>
> > **Q5.** Diversity of test cases
>
> Thank you for highlighting this critical aspect. We ensure diversity in our evaluation across multiple dimensions:
>
> 1. Modalities: Our benchmark evaluates both text-to-image (T2I) and image-to-text (I2T) models, covering multimodal input-output pairs. This ensures a comprehensive evaluation across different modalities.
>
> 2. Perspectives: MMDT includes six distinct trustworthiness perspectives: safety, hallucination, fairness, privacy, adversarial robustness, and OOD robustness. Each perspective is carefully designed to address unique aspects of MMFM vulnerabilities.
>
> 3. Scenarios: Each perspective further includes multiple challenging scenarios. For instance, the safety perspective incorporates scenarios such as vanilla harmful instructions, transformed harmful instructions, and jailbreaking harmful instructions.
>
> 4. Red-teaming Algorithms: We employ a variety of red-teaming algorithms tailored for each perspective. These include optimization-based approaches, adversarial perturbations, and semantic transformations to ensure the generated test cases are both challenging and diverse.
>
> 5. Tasks: Across modalities and perspectives, our evaluation covers a wide range of tasks such as object recognition, attribute recognition, and spatial reasoning.
>
> To illustrate this diversity, Figure 4 in the paper provides a tree taxonomy that organizes perspectives, scenarios, and tasks hierarchically, showcasing MMDT's comprehensive and structured approach.
>
>
> [1] Nemo guardrails: A toolkit for controllable and safe llm applications with programmable rails. EMNLP 2023.
>
> [2] Llama guard: Llm-based input-output safeguard for human-ai conversations. arXiv 2023.
>
> [3] Woodpecker: Hallucination correction for multimodal large language models. arXiv 2023.

---

> > ### Comment · Reviewer_SJeB · 2024-11-25
> > **Acknowledge**
> >
> > Thanks to the authors for their responses! I kept my rating unchanged since my initial rating was positive.

---

### Official Review · Reviewer_dzZE · 2024-11-02

**Soundness:** 3
**Presentation:** 3
**Contribution:** 3
**Rating:** 8
**Confidence:** 4

**Summary:**

This paper presents MMDT (Multimodal DecodingTrust), the first unified platform dedicated to comprehensively evaluating the safety and trustworthiness of multimodal foundation models (MMFMs). MMDT provides a thorough assessment of both text-to-image (T2I) and image-to-text (I2T) models across key dimensions, including safety, hallucination, fairness, privacy, adversarial robustness, and out-of-distribution generalization. Through diverse evaluation scenarios and red teaming algorithms, MMDT establishes a high-quality benchmark that uncovers significant vulnerabilities within MMFMs.

**Strengths:**

1. The evaluation dimensions in this paper are comprehensive, covering essential aspects of trustworthiness, such as safety, hallucination, fairness, privacy, adversarial robustness, and out-of-distribution robustness.
2. The paper provides a thorough evaluation of multimodal foundation models from both T2I (text-to-image) and I2T (image-to-text) perspectives, resulting in a more unified assessment.
3. The writing is well-structured and logically clear.

**Weaknesses:**

1. The formatting of the paper appears to have some issues—could it be too wide?
2. The paper lacks a description of the design of the evaluation platform. As a trustworthiness-focused evaluation platform, this section is essential to highlight its unique contributions.
3. Please provide more detail on the criteria used to select models for evaluation. Appendix A.2 offers only a brief overview, but I would like to understand the rationale behind these choices. Furthermore, the selection of evaluated models seems limited, particularly for I2T models, as classic models like the Claude series are notably missing.
4. We notice that some data in the dataset are generated using models like GPT-4, and that GPT-4o is also used as an evaluation model. Could this approach inherently favor the GPT-4 series in the evaluation?

**Questions:**

Please refer to the weakness. If questions are addressed, I will raise the rating.

---

> ### Author Response · Authors · 2024-11-22
> **Response to Reviewer dzZE (Part 1)**
>
> We thank the reviewer for their detailed feedback and for recognizing the strengths of our work, including the comprehensive evaluation dimensions and unified assessment of multimodal foundation models (MMFMs). Below, we address the specific points raised, incorporating updates to improve clarity and completeness in our revised manuscript.
>
> > **Q1.** Formatting of the paper
>
> Thank you for pointing this out. The issue stemmed from an unintended use of the “geometry” package, which affected the page layout. We have corrected this in the revised version to ensure proper formatting and presentation.
>
> > **Q2.** Lacks a description of the design of the evaluation platform
>
> We appreciate your suggestion to elaborate on the platform's design. MMDT adopts a modularized system design that balances scalability, evaluation comprehensiveness, ease of use, and extensibility. Our unified platform is designed to ensure rigorous and continuous trustworthiness evaluations for MMFMs. The platform consists of several flexible modules, enabling users to efficiently evaluate multimodal models across various perspectives.
>
> The MMDT platform is structured into the following core modules:
>
> 1. Benchmark Orchestration: This module handles data generation, data loading, and model-specific adapters (e.g., for T2I and I2T models). It supports the construction of evaluation scenarios and red teaming algorithms, as described in the main text.
>
> 1. Configuration and Job Scheduling: Users can define custom evaluation configurations, select specific perspectives or metrics, and schedule evaluation tasks for efficient execution on local or cloud resources.
>
> 1. Inference Runtimes: This module provides inference engines for efficient processing of large-scale evaluation instances. It also supports multiple cloud providers, offering cost-effective and scalable solutions.
>
> 1. Results Analysis: After evaluation, this module aggregates, analyzes, and visualizes results across perspectives, providing actionable insights into model vulnerabilities and performance.
>
> We have included a figure and a detailed discussion of the platform’s architecture in Appendix Section B of the revised paper.
>
> > **Q3.** Provide more detail on the criteria used to select models for evaluation
>
> Thank you for the suggestion. The models were chosen based on the following criteria:
>
> 1. Relevance and Popularity: Models were selected based on their adoption in research and real-world applications.
>
> 1. Coverage of Open-Source and Closed-Source Models: We included both open-source (e.g., Stable Diffusion, LLaVa) and proprietary (e.g., GPT-4V, DALL·E 3) models to ensure fair comparisons.
>
> 1. Technical Diversity: The models represent various architectures and training paradigms, providing a holistic evaluation.
>
> These criteria were aimed at ensuring a representative sample of state-of-the-art (SOTA) MMFMs for both T2I and I2T tasks. Additional details and clarifications have been added to Appendix A.2 in the revised manuscript.

---

> ### Author Response · Authors · 2024-11-22
> **Response to Reviewer dzZE (Part 2)**
>
> > **Q4.** Selection of evaluated models seems limited, particularly for I2T models (e.g., missing Claude series)
>
> Thank you for this valuable suggestion. In response to your feedback, we have expanded our evaluation to include Claude 3.5 Sonnet, a widely recognized I2T model. The updated results are shown below:
>
> | Modality | Model          | Safety   | Hallucination | Fairness | Privacy | Adv    | OOD    | Avg    |
> |----------|----------------|----------|---------------|----------|---------|--------|--------|--------|
> | T2I      | SDXL           | 69.54    | 25.33         | **63.05** | 24.79   | 54.00  | 31.15  | 44.98  |
> |          | Dreamlike      | 71.30    | 27.10         | 62.90    | 26.96   | 48.70  | 30.68  | 44.94  |
> |          | Openjourney    | 73.33    | 26.90         | 61.80    | 26.08   | 46.22  | 31.30  | 44.94  |
> |          | DF-IF          | 71.48    | 26.22         | 47.00    | 26.57   | 46.80  | 38.22  | 42.38  |
> |          | Dalle 2        | 79.07    | 27.97         | 55.00    | 32.48   | 46.66  | 33.41  | 45.93  |
> |          | Dalle 3        | **80.19** | **38.68**     | 61.75    | **36.65** | 61.38  | **58.00** | **56.78** |
> |          | FLUX           | 69.35    | 39.82         | 42.45    | 30.57   | **61.60** | 48.80  | 48.10  |
> | I2T      | LLaVa          | 36.15    | 33.07         | 94.95    | **96.09** | 70.02  | 40.32  | 61.77  |
> |          | CogVLM         | 48.29    | 37.23         | **95.65** | 85.38   | 86.50  | 51.26  | 67.39  |
> |          | Mini-InternVL  | 61.62    | 33.08         | 93.75    | 93.91   | 85.11  | 41.83  | 68.88  |
> |          | InternVL2      | 63.93    | 36.87         | 90.45    | 87.63   | 84.91  | 39.00  | 67.80  |
> |          | GPT-4V         | **98.63** | 43.75         | 79.30    | 78.23   | 85.27  | 36.85  | 70.67  |
> |          | GPT-4o         | 85.90    | **46.10**     | 80.50    | 60.76   | **90.04** | **51.68** | 69.83  |
> |          | Gemini Pro-1.5 | 78.72    | 37.07         | 84.30    | 78.16   | 83.37  | 37.06  | 66.11  |
> |          | Llama-3.2      | 83.93    | 46.15         | 97.45    | 76.98   | 84.39  | 44.66  | **72.26** |
> |          | Claude 3.5 Sonnet | 91.62  | 40.64         | 71.52    | 73.09   | 84.34  | 26.83  | 64.34  |
>
> Claude 3.5 Sonnet demonstrates robust performance in safety (91.62) and hallucination (40.64), suggesting a high ability to mitigate unsafe or inaccurate outputs. However, its performance in OOD robustness (26.83) is notably lower than many of its peers, highlighting potential challenges in handling out-of-distribution data effectively.
>
> Overall, the inclusion of Claude 3.5 Sonnet provides valuable insights into a different model family and highlights the diversity of approaches in multimodal foundation models. We believe this addition strengthens the comprehensiveness of our evaluation and enables a more complete comparison of state-of-the-art I2T models.
>
> > **Q5.** Could the use of GPT-4 in dataset generation and evaluation inherently favor GPT-4 series models?
>
> For vanilla harmful instructions in the safety perspective, we curate our own dataset by prompting GPT while incorporating human inspection in the loop. The harmful instructions are inspected by the authors directly. We did not filter the data based on whether they will be rejected by GPT models. Besides, using GPT as an evaluation model is widely adopted in LLM safety evaluation, such as [1, 2]. Therefore, we believe our data curation and evaluation process is fair for the target models.
>
> [1] Liu, Xiaogeng, et al. "Autodan: Generating stealthy jailbreak prompts on aligned large language models." arXiv preprint arXiv:2310.04451 (2023).
>
> [2] Xu, Zhangchen, et al. "Safedecoding: Defending against jailbreak attacks via safety-aware decoding." arXiv preprint arXiv:2402.08983 (2024).

---

> > ### Comment · Reviewer_dzZE · 2024-11-25
> >
> > Thank you for your response. I decide to keep my rating as I am still confused about whether the data generated by the GPT model is actually beneficial for the GPT model itself. Perhaps you could consider incorporating some other commercial models to generate data and verify the performance of the GPT model in this regard. The current explanation does not convince me.

---

> ### Author Response · Authors · 2024-11-29
>
> Thank you for your constructive suggestion. Following your advice, we have generated new data using an alternative model to further validate our findings. Since our benchmark already evaluates most popular commercial models, we selected **QWEN-2.5-72B-Instruct**, a leading open-source model that is recent, highly capable, and not included in our evaluated models, to generate new data. This choice avoids overlap with the models evaluated in our benchmark while maintaining high data quality. Specifically, we regenerated the dataset for the safety perspective in the harmful intention hidden in illustration scenario.
>
> We then evaluate three commercial models (GPT-4o-2024-05-13, Gemini-pro-1.5-001, and Claude-3.5-Sonnet-20241022) and the largest open model (Llama-3.2-90B-Vision-Instruct) on this newly generated dataset. To ensure unbiased evaluation, we employ the same three commercial models as judges and use a majority vote across their judgments as the final metric. The evaluation results on the **QWEN-generated dataset** are presented in **Table 1**, while detailed results for the previously **GPT-4o-generated dataset** are provided in **Table 2**.
>
> Key Findings:
>
> 1. Performance Comparison: Models usually perform slightly better on the GPT-4o-generated dataset. For instance, Llama-3.2-90B improves by 3.85%, GPT-4o by 3.84%, and Gemini-pro by 2.05%.
>
> 1. Judgment Consistency: Evaluation results are consistent across the different judge models. The majority vote further improves stability, showing consensus on the ranking: GPT-4o < Llama-3.2-90B < Gemini-pro ≈ Claude-3-5. The ranking holds across datasets generated by both QWEN and GPT-4o.
>
> These results indicate that while datasets generated using different models may introduce slight variations, they do not fundamentally alter the evaluation outcomes or rankings. To address concerns about dataset diversity and potential bias, we will further review the newly generated Qwen dataset carefully and include the dataset in our revision. Furthermore, we will employ three different judge models in the evaluation platform to ensure a more comprehensive and unbiased assessment.
>
> We sincerely appreciate your suggestion to diversify the datasets and evaluation models. We believe these enhancements will contribute to a more comprehensive and fair evaluation. We hope this updated analysis and inclusion address your concerns. Thank you again for your valuable feedback.
>
>
> Table 1: Evaluation using the new QWEN-generated dataset
>
> | Judge model / Evaluated model   | Llama-3.2-90B-Vision-Instruct | GPT-4o-2024-05-13 | Gemini-pro-1.5-001 | claude-3-5-sonnet-20241022 |
> |---------------------------------|------------------------------|--------------------|---------------------|---------------------------|
> | GPT-4o-2024-05-13              | 85.90                        | 83.08             | 92.05              | 93.59                    |
> | Gemini-pro-1.5-001             | 83.59                        | 75.38             | 89.23              | 92.05                    |
> | claude-3-5-sonnet-20241022     | 84.10                        | 76.92             | 91.54              | 93.33                    |
> | Majority vote                  | 84.36                        | 77.44             | 91.28              | 93.98                    |
>
> Table 2: Evaluation using the original GPT-4o-generated dataset
>
> | Judge model / Evaluated model       | Llama-3.2-90B-Vision-Instruct | GPT-4o-2024-05-13 | Gemini-pro-1.5-001 | claude-3-5-sonnet-20241022 |
> |-------------------------------------|-------------------------------|--------------------|---------------------|---------------------------|
> | GPT-4o-2024-05-13                  | 89.49                         | 84.62              | 94.62               | 88.97                     |
> | Gemini-pro-1.5-001                 | 86.41                         | 79.23              | 93.33               | 93.08                     |
> | claude-3-5-sonnet-20241022         | 89.49                         | 81.03              | 92.31               | 95.38                     |
> | Majority vote                      | 88.21                         | 81.28              | 93.33               | 93.08                     |

---

> > ### Author Response · Authors · 2024-12-01
> >
> > Dear Reviewer dzZE,
> >
> > Thank you for taking the time and effort to review our submission. We greatly appreciate your thoughtful feedback, which has been instrumental in helping us improve our work.
> >
> > We have carefully addressed the concerns you raised and provided detailed responses, including conducting additional experiments based on your suggestions. We wanted to kindly follow up to confirm whether our responses have adequately addressed your concerns or if there are any remaining points we could further clarify or improve, especially as tomorrow is the final day for the discussion period.
> >
> > Thank you once again for your invaluable feedback and contributions to the review process!

---

> > > ### Comment · Reviewer_dzZE · 2024-12-03
> > >
> > > Thanks for authors' efforts. My concerns have been addressed and thus, I raise the rating.

---

> > > > ### Author Response · Authors · 2024-12-03
> > > >
> > > > Thank you for your response! We would like to thank you again for the valuable suggestions and comments in the review process!

---

### Official Review · Reviewer_AKat · 2024-11-04

**Soundness:** 3
**Presentation:** 4
**Contribution:** 3
**Rating:** 8
**Confidence:** 3

**Summary:**

This paper introduces MMDT (Multimodal DecodingTrust), the first unified platform for comprehensive safety and trustworthiness evaluation of multimodal foundation models (MMFMs). Unlike existing benchmarks that primarily assess helpfulness or focus on limited aspects such as fairness and privacy, MMDT evaluates MMFMs across multiple dimensions: safety, hallucination, fairness/bias, privacy, adversarial robustness, and out-of-distribution generalization. The platform includes diverse evaluation scenarios and red teaming algorithms to create a challenging benchmark. Through testing various multimodal models, MMDT identifies vulnerabilities and areas for improvement, contributing to the development of safer and more reliable MMFMs and systems.

**Strengths:**

1. MMDT provides a thorough evaluation across multiple dimensions of trustworthiness—safety, hallucination, fairness, privacy, adversarial robustness, and out-of-distribution (OOD) generalization. This comprehensive approach is unique and valuable for identifying vulnerabilities in multimodal foundation models (MMFMs).'
2. The study includes a range of scenarios tailored to different tasks, such as object recognition, counting, spatial reasoning, and attribute recognition. The use of red teaming algorithms for generating challenging data is innovative and enhances the robustness of the benchmark​.
3. By testing MMFMs in applications like autonomous driving and healthcare, the paper addresses real-world scenarios where safety and reliability are critical. This relevance adds practical value to the findings.
4. I like the very detailed appendix, which provide comprehensive information about the trustworthiness of MMFMs.

**Weaknesses:**

1. Although the paper identifies vulnerabilities and areas for improvement, it lacks a detailed discussion or guidance on specific mitigation techniques for enhancing model trustworthiness. Including concrete strategies for addressing the identified issues could improve the paper’s utility.
2. While MMDT identifies vulnerabilities across multiple dimensions, it may lack sufficient guidance on interpreting and addressing these vulnerabilities for practitioners, particularly those unfamiliar with the nuances of each trustworthiness perspective.
3. The study highlights general issues with MMFMs, but not propose in-depth comparisons between models and answer what make specific model has  strength and weaknesse.​

**Questions:**

Thank you for submitting your work to ICLR! Overall, I like to read this paper as it produces very comprehensive evaluation in terms of the trustworthiness of the MMFMs. While reading this work, I have the following questions and I wish the author can resolve them:

1. In Table 1, as jailbreaking harmful instructions are optimizing adversarial prompts to help bypass safety filters, why the jailbreaking has lower HGR compared with the some of Vanilla case? I thought the jailbreaking is optimizing adversarial prompts, so that it should achieve higher success rate with higher HGR and BR, but it seems that the HGR is not optimized. Can the author explain why they end with this result, and why not optimizing HGR when crafting the adversarial prompts?

2. It seems like GPT-4v achieves best performance in defending against generating harmful text output with different image manipulation, can the author discuss the possible reason behind this?

3. You extend beyond social stereotypes to include decision-making and “overkill fairness.” Could you clarify the criteria used to define and measure overkill fairness? For example, how do you balance fairness goals with historical accuracy, and do you find certain models more prone to overkill fairness in specific contexts?

4. In your OOD evaluation, you find that DALL·E 3 and GPT-4o outperform other models, especially in tasks involving spatial reasoning and attribute recognition. Could you specify the types of OOD transformations that most challenge MMFMs, and whether these models employ specific techniques that enhance their resilience to such transformations?

5. The privacy benchmark assesses object-level memorization, focusing on CLIP embedding similarity to detect privacy leaks. Given this method, could you discuss any limitations or potential improvements? For instance, could certain model fine-tuning approaches mitigate memorization without significantly impacting performance?

---

> ### Author Response · Authors · 2024-11-22
> **Response to Reviewer AKat (Part 1)**
>
> We sincerely thank the reviewer for the thoughtful feedback and for recognizing the strengths of our work, including the comprehensive evaluation of MMFMs, the innovative use of red teaming algorithms, and the practical relevance of our findings to real-world applications. Below, we address the specific weaknesses and questions raised, and outline the improvements we will incorporate into our revised paper.
>
> > **Q1.** Lacks a detailed discussion or guidance on specific mitigation techniques for enhancing model trustworthiness.
>
> Thank you for your valuable suggestion. We have expanded our discussion of mitigation strategies for each trustworthiness perspective, detailed in Appendix C.3 of the revised paper. Key strategies include:
>
>  - Safety: Given that safety risks persist across all target models according to our evaluation, we recommend implementing more advanced mitigation strategies at various stages. During the training stage, utilizing Reinforcement Learning from Human Feedback (RLHF) with high-quality data based on a comprehensive taxonomy of risk categories is essential to reduce the risk of generating unsafe content. During the deployment stage, input and output-level guardrails, such as [1] and [2], could be adopted to detect and filter out unsafe content provided by users or generated by the model. Additionally, developing certified defenses against jailbreaking attacks can further mitigate these risks.
>
>  - Hallucination: Our findings in MMDT suggest that MMFMs tend to hallucinate primarily due to: 1) poor visual grounding; 2) imbalanced attention between textual and visual information; and 3) poor reasoning or instruction-following abilities. To address these issues, we can consider the following mitigation strategies: 1) utilize external tools to enhance visual grounding, as indicated in [3]; 2) adaptively calibrate attention to ensure a balanced focus on both textual and visual tokens; 3) employ supervised fine-tuning or preference tuning to reduce hallucination during training; and 4) leveraging external knowledge base for factual image retrieval to mitigate hallucination for text-to-image generation.
>
>  - Fairness: Since image generation models exhibit more severe bias issues compared to text generation models based on our observations, we advocate for the development of more effective bias mitigation strategies in the image domain. Addressing bias in image generation models is particularly challenging due to their increased complexity and the absence of automatic reward feedback. Therefore, leveraging human preference annotations and techniques such as DPO to enforce fairness in text-to-image models presents a promising direction for future work.
>
>  - Privacy: Based on our evaluation results, we recommend employing privacy-preserving techniques during both the training and inference stages for MMFMs. During training, using differentially private learning algorithms or differentially private synthetic multimodal data could help alleviate concerns about privacy leakage. For inference, we suggest implementing scrubbing or anonymization on user-provided images and text to remove sensitive attributes. Additionally, MMFMs could incorporate privacy-aware instruction tuning and reject queries related to sensitive human attributes.
>
>  - Adversarial Robustness: According to our findings that adversarial examples generated by GCG and MMP has high transferability to other black-box models, we recommend to employ these algorithms to attack a wide range of open models to collect challenging data and mix the data into the training blend. These adversarial datasets will help improve model robustness vie trustworthiness fine-tuning.
>
>  - OOD Robustness: Given that superior in-distribution performance in MMDT typically leads to better out-of-distribution performance, we recommend further enhancing the models’ benign performance by increasing the training dataset quality and diversity. Additionally, collecting a diverse training dataset with various styles through data augmentation and incorporating diverse tasks, such as spatial reasoning and attribute recognition, can potentially improve the robustness of multi-modal models against different styles and tasks. Furthermore, we recommend multi-modal models to incorporate “I don’t know” option during training, enabling them to perform OOD detection by themselves rather than generating random answers.

---

> ### Author Response · Authors · 2024-11-22
> **Response to Reviewer AKat (Part 2)**
>
> > **Q2.** Lacks sufficient guidance on interpreting vulnerabilities identified by MMDT.
>
> Thank you for your suggestion. We have added detailed explanations of vulnerabilities for each trustworthiness perspective in Appendix C.2 to guide practitioners. Highlights include:
>
> - Safety: Safety risks often emerge from insufficient coverage of risky scenarios during alignment and inadequate mechanisms to filter unsafe outputs. Many models lack fine-grained, multi-level moderation systems, leading to vulnerabilities such as generating inappropriate or harmful content. These issues are exacerbated in scenarios like jailbreaking, where adversarial prompts can exploit model weaknesses.
>
> - Hallucination: Hallucinations are primarily caused by weak grounding in visual and textual information, unbalanced attention mechanisms, and limited reasoning capabilities. For instance, in MMDT we find models may generate outputs that misrepresent relationships between image content and textual prompts due to incomplete multimodal understanding.
>
> - Fairness: The fairness issue primarily arises from biases inherent in the training data. These biases can not only persist but also be amplified [4], potentially deviating significantly from the original training-data statistics. While alignment efforts aim to mitigate bias, MMDT still observes numerous failure modes across diverse domains, as achieving distribution-level alignment presents substantial challenges.
>
> - Privacy: Privacy vulnerabilities are linked to the inadvertent memorization of sensitive information during training. Models trained on large datasets that include private data may inadvertently expose identifiable information, highlighting the need for privacy-preserving training techniques.
>
> - Adversarial robustness: During training, models are typically exposed only to clean data, which may not comprehensively cover all relevant variations or edge cases. They are not trained to handle perturbed or adversarial inputs, leaving them unprepared for adversarial scenarios.
>
> - Out-of-distribution robustness: OOD vulnerabilities arise from limited coverage of diverse styles, tasks, or domains in the training data. This results in models that perform well on in-distribution data but fail to generalize to novel scenarios, such as rare visual styles or linguistic constructs.
>
> > **Q3.** Lack of in-depth comparisons between models to explain strengths and weaknesses.
>
> Thank you for your comment. In our evaluations, we observed the following:
>
> 1. Conservativeness: Closed-source models like GPT-4V tend to be more conservative, which contributes to their superior performance in safety benchmarks but may hinder their creativity.
>
> 1. Alignment: Closed-source models exhibit better alignment with safety principles due to rigorous alignment fine-tuning, making them harder to jailbreak compared to open-source models.
>
> 1. Architecture and Scale: Larger-scale models such as GPT-4V exhibit better performance across multiple perspectives, particularly in safety and hallucination, likely due to their extensive pretraining and sophisticated architectures. On the other hand, smaller models like LLaVa often struggle with these tasks.
>
> 1. Training Data Diversity: Models trained on diverse and curated datasets, such as DALL·E 3, tend to perform better in OOD robustness and hallucination metrics. In contrast, open-source models, such as Stable Diffusion variants, often rely on less curated datasets, leading to weaker performance.
>
> We also add these observations in Appendix C.1 to offer practitioners clearer comparisons and insights into the strengths and weaknesses of each model.
>
> > **Q4.** Jailbreaking prompts have lower HGR compared to Vanilla cases.
>
> This is due to the design of our jailbreaking algorithms, which prioritize maximizing the bypass rate (BR) on a surrogate model. While the adversarial prompts bypass safety filters, the generated content is not necessarily highly harmful, resulting in lower Harmful Generation Rate (HGR) in certain cases. For example, some jailbreak prompts elicit irrelevant or benign outputs instead of harmful content.
>
> > **Q5.** GPT-4V’s superior performance against harmful text output with image manipulation.
>
> GPT-4V achieves strong performance due to its fine-tuning for multimodal alignment, which emphasizes safe and accurate responses to complex multimodal inputs. By contrast, open-source models exhibit weaker defenses due to limited multimodal alignment during training.

---

> ### Author Response · Authors · 2024-11-22
> **Response to Reviewer AKat (Part 3)**
>
> > **Q6.** Overkill fairness: Criteria and model tendencies.
>
> Thank you for your question about overkill fairness. We define overkill fairness as when models sacrifice historical/factual accuracy in pursuit of fairness goals. For instance, models might generate images of Black people when prompted about "the Founding Fathers" –- while diversity is important, this is NOT what we want. To address this, we included overkill fairness in our evaluation scope, measuring whether models can recognize and maintain historical facts (e.g., Figure 15 and the blue box on page 45 in Appendix). Our findings indicate that all tested models struggle significantly with overkill fairness. We believe this issue persists because existing fairness benchmarks focus solely on measuring diversity/fairness without considering potential side effects. Ideal models should promote diversity “appropriately” rather than indiscriminately across all contexts. Our paper identifies a critical tradeoff between fairness and historical/factual accuracy, introducing a challenging balance that the research community needs to address. This finding represents a significant contribution to the field.
>
> > **Q7.** OOD transformations and resilience.
>
> Thank you for your insightful question! We provide the detailed performance of each task and specific transformations in **Table 40** in Appendix G.1 for T2I models and **Table 43** in Appendix G.2 for I2T models. Below are the key takeaways for OOD transformations:
>
> For T2I OOD Transformations
>
> - **Shakespearean styles** generally cause greater challenges for tasks such as helpfulness and counting, while causing similar performance drops to rare linguistic structures in tasks like spatial reasoning and attribute recognition.
>
> For I2T OOD Transformations
>
> - **Zoom Blur** corruptions and **Van Gogh style** transformations result in the most significant performance drops. Zoom Blur corruptions degrade performance by more than 25% across most models, with the exception of CogVLM. Similarly, Van Gogh style transformations lead to more than 25% performance drops for most models, except CogVLM, GPT-4-o, and Gemini Pro-1.5.
>
> Since the technical details of closed-source models such as DALL·E 3 and GPT-4-o have not been released, we cannot contend whether they incorporate additional augmentations or techniques to improve resilience. However, none of the open-source models with available technical reports mention the use of additional augmentations/techniques to address severe corruptions in I2T tasks or diverse style transformations in both I2T and T2I models.
>
>
> > **Q8.** Privacy benchmark: Limitations and improvements.
>
> Thanks for the thoughtful comment.
> We acknowledged the limitations of CLIP embedding similarity as a measurement, which may not fully capture nuanced privacy leaks. The effectiveness of this method hinges on the training data and objective of CLIP. If CLIP's embedding space does not adequately represent certain types of sensitive information, the benchmark might underperform.
>
> As mitigation of memorization, using differentially private learning algorithms or differentially private synthetic multimodal data during training/fine-tuning could help alleviate concerns about privacy leakage.
>
>
> [1] Nemo guardrails: A toolkit for controllable and safe llm applications with programmable rails. EMNLP 2023.
>
> [2] Llama guard: Llm-based input-output safeguard for human-ai conversations. arXiv 2023.
>
> [3] Woodpecker: Hallucination correction for multimodal large language models. arXiv 2023.
>
> [4] A systematic study of bias amplification. arXiv 2022.

---

### Author Response · Authors · 2024-11-22
**General Response**

We sincerely thank the reviewers for their constructive feedback and thoughtful questions about our paper. We greatly appreciate the reviewers’ recognition of our contributions, including the comprehensive evaluation framework, innovative construction of red teaming algorithms, and practical relevance for improving the trustworthiness of MMFMs. Below, we outline the key changes we have made in response to the reviewers’ comments and summarize how their feedback has helped us strengthen the paper. The paper is updated and the revision is highlighted in blue.

1. Additional model evaluation. We have added the evaluation of additional models such as Claude 3.5 Sonnet, further enhancing the comprehensiveness of our benchmark.

1. Model Selection Criteria. We have clarified the criteria for selecting models for evaluation in Appendix A.2, ensuring transparency about the choices made.

1. Model Comparisons. We have added a thorough discussion in the main paper and Appendix C.1 to highlight the strengths and weaknesses of different models, including observations on conservativeness, alignment, architecture, and training data diversity.

1. Interpretation of Vulnerabilities. To provide clearer guidance on interpreting vulnerabilities, we now include detailed explanations for each trustworthiness perspective in Appendix C.2, outlining the causes and implications of the observed issues in MMFMs.

1. Mitigation Strategies for Enhancing Trustworthiness. We have expanded the discussion on mitigation strategies for each trustworthiness perspective (safety, hallucination, fairness, privacy, adversarial robustness, and out-of-distribution robustness) in the main paper with more details in Appendix C.3, as suggested. These strategies provide actionable insights for practitioners to address the vulnerabilities identified by MMDT.

1. Evaluation Platform Design. To address requests for more detail, we now include a description of the platform’s modular design in the main paper, along with more details in Appendix B, along with a figure illustrating its architecture. This highlights the platform's scalability, extensibility, and ability to conduct rigorous multimodal evaluations.

1. Unique Findings in MMFMs. We have emphasized discoveries unique to MMFMs compared to LLMs, such as cross-modal vulnerabilities, attribute amplification, and modality-specific privacy risks, addressing concerns about limited analytical insights.

In summary, we have taken significant steps to address the reviewers' questions by expanding our evaluations, improving clarity, enhancing the structure of our paper, and adding discussions on practical mitigation strategies. We also integrated these improvements in our revisions and we believe they will further strengthen the contribution and impact of our work.

---

### Meta-Review · Area_Chair_gS6m · 2024-12-20

**Metareview:**

This paper introduces MMDT (Multimodal DecodingTrust), the first unified platform for comprehensive safety and trustworthiness evaluation of multimodal foundation models (MMFMs). Unlike existing benchmarks that primarily assess helpfulness or focus on limited aspects such as fairness and privacy, MMDT evaluates MMFMs across multiple dimensions: safety, hallucination, fairness/bias, privacy, adversarial robustness, and out-of-distribution generalization. The platform includes diverse evaluation scenarios and red teaming algorithms to create a challenging benchmark. Through testing various multimodal models, MMDT identifies vulnerabilities and areas for improvement, contributing to the development of safer and more reliable MMFMs and systems. The reviewers agree on the work: (1) MMDT provides a thorough evaluation across multiple dimensions of trustworthiness. This comprehensive approach is unique and valuable for identifying vulnerabilities in MMDT. (2) The study covers a range of scenarios and tasks including the real-world scenarios where safety and reliability are critical. (3)  The writing is well-structured and logically clear with a detailed appendix.

The reviewers also have some concerns including: (1) lacking a detailed discussion or guidance on specific mitigation techniques for enhancing model trustworthiness. (2) Lacking in-depth comparisons between models and answers to what makes specific models have strengths and weaknesses.​ (3) Lacking a description of the design of the evaluation platform. (4) Lacking the explanation of the use of GPT-4 in dataset generation and evaluation inherently favors GPT-4 series models. (5) The lack of analytical components or theoretical insights for the MMFMs vs. LLMs. Most of the issues are properly addressed in the rebuttal and revised version.

According to the reviewers' comments and rebuttal discussion, we agree this work is important for the community and should be definitely accepted.

**Additional Comments On Reviewer Discussion:**

All reviewers provide solid comments. Reviewer AKat indicates the work provides a thorough evaluation across multiple dimensions of trustworthiness. This comprehensive approach is unique and valuable for identifying vulnerabilities in MMDT, but points out some lack of discussions. Reviewer dzZE agrees that the evaluation dimensions in this paper are comprehensive, but points out the lack of a description of the design of the evaluation platform. Reviewer SJeB and Reviewer mkAr also agree the work provides a comprehensive evaluation framework to evaluate Multimodal Foundation Models (MMFMs) and concerned about the difference in the evaluation of the LLMs. The rebuttal addresses most of the concerns of reviewers and revises the submission.

---

### Decision · Program_Chairs · 2025-01-22

Accept (Poster)